# Comparison of Circulating Cell-Free DNA Extraction Methods for Downstream Analysis in Cancer Patients

**DOI:** 10.3390/cancers12051222

**Published:** 2020-05-13

**Authors:** Paul van der Leest, Pieter A. Boonstra, Arja ter Elst, Léon C. van Kempen, Marco Tibbesma, Jill Koopmans, Anneke Miedema, Menno Tamminga, Harry J. M. Groen, Anna K. L. Reyners, Ed Schuuring

**Affiliations:** 1Department of Pathology and Medical Biology, University of Groningen, University Medical Center Groningen, Hanzeplein 1, 9713 GZ Groningen, The Netherlands; p.van.der.leest@umcg.nl (P.v.d.L.); a.ter.elst@umcg.nl (A.t.E.); l.van.kempen@umcg.nl (L.C.v.K.); marcotibbesma4@hotmail.com (M.T.); jillkoopmans@hotmail.com (J.K.); a.miedema@umcg.nl (A.M.); 2Department of Medical Oncology, University of Groningen, University Medical Center Groningen, Hanzeplein 1, 9713 GZ Groningen, The Netherlands; p.a.boonstra@umcg.nl (P.A.B.); a.k.l.reyners@umcg.nl (A.K.L.R.); 3Department of Pulmonary Diseases, University of Groningen, University Medical Center Groningen, Hanzeplein 1, 9713 GZ Groningen, The Netherlands; m.tamminga@umcg.nl (M.T.); h.j.m.groen@umcg.nl (H.J.M.G.)

**Keywords:** ccfDNA, ctDNA, extraction, yield, integrity

## Abstract

Circulating cell-free DNA (ccfDNA) may contain DNA originating from the tumor in plasma of cancer patients (ctDNA) and enables noninvasive cancer diagnosis, treatment predictive testing, and response monitoring. A recent multicenter evaluation of workflows by the CANCER-ID consortium using artificial spiked-in plasma showed significant differences and consequently the importance of carefully selecting ccfDNA extraction methods. Here, the quantity and integrity of extracted ccfDNA from the plasma of cancer patients were assessed. Twenty-one cancer patient-derived cell-free plasma samples were selected to compare the Qiagen CNA, Maxwell RSC ccfDNA plasma, and Zymo manual quick ccfDNA kit. High-volume citrate plasma samples collected by diagnostic leukapheresis from six cancer patients were used to compare the Qiagen CNA (2 mL) and QIAamp MinElute ccfDNA kit (8 mL). This study revealed similar integrity and similar levels of amplified short-sized fragments and tumor-specific mutants comparing the CNA and RSC kits. However, the CNA kit consistently showed the highest yield of ccfDNA and short-sized fragments, while the RSC and ME kits showed higher variant allelic frequencies (VAFs). Our study pinpoints the importance of standardizing preanalytical conditions as well as consensus on defining the input of ccfDNA to accurately detect ctDNA and be able to compare results in a clinical routine practice, within and between clinical studies.

## 1. Introduction

Extensive research has been performed to utilize blood-based analytes for detection and monitoring of the disease in cancer patients. In many bodily fluids, including blood plasma, circulating cell-free DNA (ccfDNA) is present [1]. In healthy individuals, ccfDNA originates primarily from cell degradation through apoptosis or necrosis of cells of the hematopoietic lineage, resulting in shedding of genomic DNA into the circulation [2,3]. CcfDNA released through apoptosis consists of short fragments (<1000 bp), while ccfDNA shedded into circulation by exosomes or necrotic (tumor) cells is longer in size (>1000 bp) [4,5]. 

The application of ccfDNA for molecular profiling has recently taken a flight in the field of oncology. Elevated levels of ccfDNA can be detected in cancer patients of which a small fraction of the ccfDNA originates from tumor cells [6]. This so-called circulating tumor DNA (ctDNA) reflects the molecular characteristics of tumor tissue and promises to negate the limitations of conventional tissue biopsy (e.g., invasiveness, accessibility, and heterogeneity) [7]. Similar to ccfDNA, ctDNA is shedded into the bloodstream through apoptosis, necrosis, or active excretion [8]. The majority of ctDNA in plasma has an apoptotic origin and in general their size corresponds to nucleosome-protected DNA, which ranges from 120 to 220 base pairs (bp) that peaks around 167 bp [9]. In addition to the apoptotic short-sized fragments, the ccfDNA in cell-free plasma also contains very long-sized fragments (~10,000 bp) resulting from (tumor) cell necrosis or derived from hemolysis during blood withdrawal and processing [10,11,12]. Altogether, the actual ctDNA fraction mostly represented in the shorter-sized fragments is often less than 1% of the total ccfDNA [5,7]. Thus, the extracted ctDNA is highly fragmented and has a short half-life, which can complicate subsequent analyses [1]. Therefore, efficient extraction of the short-sized fragments and highly sensitive techniques are required to detect these low abundant ctDNA fragments [13]. Nonetheless, ctDNA contains tumor-specific mutations that can be detected in the plasma using highly sensitive techniques such as droplet digital polymerase chain reaction (ddPCR) and highly-sensitive next-generation sequencing (NGS) [14,15]. The detection of these short-sized ctDNA fragments in the plasma enables the (early) detection of new or recurrent predictive cancer biomarkers and is applicable for monitoring treatment response using a minimally invasive strategy [16]. 

With the increasing interest for ccfDNA-based diagnostics, the number of ccfDNA extraction kits and methodologies has expanded drastically in recent years [17]. There is a great variety regarding the extraction method, plasma input, throughput, and price of the kits. To compare the performance of various ccfDNA extraction kits, total DNA yield is generally used as an outcome parameter. However, a substantial part of the ccfDNA originates from genomic DNA fragments of nontumor tissue released during blood withdrawal and processing of the samples due to hemolysis [18]. These increased levels of larger DNA fragments can interfere with the sensitivity of ctDNA detection and could result in false negativity. In addition, no methods exist that enrich short-sized ccfDNA fragments or are able to discriminate ctDNA from regular ccfDNA. Altogether, it is important to evaluate the preanalytical conditions and integrity of the extracted ccfDNA when using quantitative approaches to accurately detect mutants during diagnostics or monitoring of the disease. 

Many studies reported the comparison of different ccfDNA extraction methods [19,20,21,22,23,24,25,26]. At present, however, no collective international standardized protocols are available regarding the preanalytical ccfDNA extraction conditions. Most ccfDNA comparison studies were performed using reference samples, that typically consist of either artificial plasma or pooled plasma samples from healthy individuals both spiked with purified DNA [19,20,21,22]. Furthermore, most ccfDNA extraction methods do not provide any information regarding the preanalytical conditions of the obtained DNA [17,27]. A recent multicenter evaluation of workflows by the CANCER-ID consortium revealed considerable differences between various ccfDNA extraction methods regarding the quantity and integrity of extracted ccfDNA using artificial spiked-in plasma and showed the relevance of carefully selecting extraction methods and considering preanalytical conditions of the extracted ccfDNA [7]. In line with this CANCER-ID study, the aim was to evaluate the quantity and integrity of extracted ccfDNA from cancer patient-derived plasma samples using different ccfDNA extraction kits. Plasma samples from patients with either a gastrointestinal stromal tumor (GIST) or nonsmall cell lung carcinoma (NSCLC) were selected for the comparison of three different plasma ccfDNA extraction techniques (QIAamp Circulating Nucleic Acid Kit (CNA), Maxwell RSC ccfDNA Plasma Kit (RSC), and Zymo Quick ccfDNA Serum & Plasma Kit (Z)). 

In the clinical setting, there is a rising demand for processing higher volumes of plasma in one run to minimize expenditure and generate highly concentrated eluates to enable subsequent analyses for diagnostic purposes such as NGS [28,29]. The commonly used CNA kit applying 2 mL of plasma served in our study as a reference to compare ccfDNA extraction with the QIAamp MinElute ccfDNA kit (ME) that preferably enables ccfDNA extraction from 8 mL of plasma. For this purpose, we collected a unique set of high-volume citrate plasma samples collected by diagnostic leukapheresis (DLA) from six patients to evaluate the recovery of ccfDNA using 8 mL of plasma, which is an average amount collected from two blood collection tubes (BCTs) compared to the CNA kit using 2 mL of plasma. 

## 2. Results

### 2.1. Selection of Plasma Samples from Cancer Patients

Twenty-one plasma samples from eighteen cancer patients with metastatic disease were selected based on plasma availability in our plasma Biobank sufficient to be able to perform three ccfDNA extractions on the same sample (4 mL in total). Eight samples from seven GIST patients and thirteen samples from eleven patients with NSCLC were used (Appendix A). 

### 2.2. Quantitative Comparison of DNA Yield with Different ccfDNA Extraction Methods

CcfDNA extraction using three different ccfDNA extraction kits showed a broad range of concentrations when measured with Qubit, varying from 1.53 ng ccfDNA per mL of plasma to 110 ng/mL (Figure 1A,B). Overall, ccfDNA extraction using the CNA kit resulted in a significantly higher yield compared to the RSC (*p* < 0.001) and Z kits (*p* < 0.01) (Figure 1A). Extraction using the CNA kit (Qiagen, Hilden, Germany)yielded the highest levels in eighteen out of twenty-one samples, whereas the lowest amount of ccfDNA was obtained in fourteen samples with the RSC kit (Figure 1B).

### 2.3. CcfDNA Integrity and Mutation Detection Assessment

In order to validate the various fragment sizes in ccfDNA, a sample analysis using the Fragment Analyzer revealed no significant differences in the short-to-medium-sized fragment ratios (Appendix A). In order to determine the amplifiability assessment of fragment sizes, using the β-actin one-tube 3-size ddPCR assay resulted in a significantly higher number of copies per mL of plasma for the 137 and 420 bp fragment lengths in ccfDNA extracted with the CNA kit compared to both the RSC (*p* < 0.05 and *p* < 0.01, respectively) and Z kits (*p* < 0.0001 and *p* < 0.001, respectively) (Figure 2A,B). For the long 1950 bp fragments, the CNA kit only extracted significantly more compared with the RSC kit (Figure 2C). All median values and interquartile ranges are depicted in Appendix A. Extracted ccfDNA revealed no significantly different 137/420 bp fragment ratios, which is in agreement with the results from the Fragment Analyzer (Appendix A). The 137/1950 bp fragment ratios only showed a significant increase in the extraction of long-sized ccfDNA fragments with the Z kit compared with the RSC kit (Appendix A). When comparing the number of copies of the 137 bp fragment per ccfDNA input (in ng), the mean number of copies per ng ccfDNA was similar between the CNA and RSC kits (*p* = 0.247) and also for each separate plasma sample using the CNA or RSC kit, no concordant pattern was observed (Figure 2D).

Next, to compare the yield of the ctDNA fraction, mutation-specific ddPCR assays were performed on the extracted ccfDNA of multiple patients of which in four plasmas the mutations were detectable (Appendix A). Interestingly, ccfDNA extraction using the CNA kit resulted in more mutant copies per mL of plasma in two cases, while RSC-extracted ccfDNA showed more mutant copies per mL of plasma in the other two cases (Figure 3A). In Figure 3B, the detected number of mutant copies is plotted against the ccfDNA input in ng. Hereby, it can be determined whether the input amount affects the detected number of mutant copies per mL of plasma. The detected discrepancies are irrespective of the input amount of ccfDNA (Figure 3B). In regard to variant allelic frequency (VAF), in three of the four samples RSC-extracted ccfDNA displayed a higher VAF compared with CNA-extracted ccfDNA (Figure 3C). 

Overall, this data using twenty-one different plasma samples of eighteen patients with cancer revealed a larger total yield of ccfDNA and a relatively higher number of short-sized (137 and 420 bp) fragments when using the CNA kit compared to the RSC kit, which is in agreement with data observed in spiked-in samples [7]. However, both methods provided the same number of short-sized fragments (*n* = 21) and mutant copies (*n* = 4) relative to the amount of ccfDNA input in ng.

### 2.4. Comparing ccfDNA Extraction Kits Using High-Volume Citrate Plasma Samples

The extracted ccfDNA is generally used for subsequent mutational analysis techniques such as ddPCR and NGS, which mostly require high quantities of ccfDNA derived from preferably 4 mL of plasma [28,29]. Therefore, processing high volumes of plasma in a single reaction is preferred. For this purpose, we compared the magnetic beads-based ME kit (8 mL plasma) specifically designed to process high volumes of plasma with the most frequently used ccfDNA CNA extraction kit (2 mL) (see Table 1). In this analysis, ccfDNA extracted with the ME kit was compared with the CNA kit for yield, integrity, and amplifiability of short-sized fragments. 

DLA samples from six NSCLC patients with established mutations were selected based on plasma availability (>10 mL). With respect to yield, extraction using the CNA kit resulted in a 3-fold more ccfDNA per mL of plasma compared to the ME kit (Figure 4A). Each CNA-extracted sample had a higher yield than the paired ME-extracted samples (Figure 4B). A sample analysis with the Fragment Analyzer showed enrichment of short-sized fragments in ccfDNA extracted with the ME kit compared to the CNA kit, as demonstrated by an increased number of 50–250 bp fragments and a higher short-to-medium-sized fragments ratio (Table 2).

The amplifiability of ccfDNA was assessed using the β-actin one-tube 3-size ddPCR. The median number of copies per mL of plasma of all three fragment lengths (137, 420, and 1950 bp) of the ME kit was slightly lower but not significantly different compared to the CNA kit (Figure 5A–C). Extracted ccfDNA from the ME and CNA kits revealed no significant differences in the 137/420 bp and 137/1950 bp ratios, whereas the ME kit revealed a significantly higher short-to-medium-size ratio on the Fragment Analyzer (Table 2). When considering the input amount of ccfDNA, a strong increase of 137 bp copies per ng ccfDNA (Figure 5D) as well as for the 420 and 1950 bp fragment lengths (Appendix A) was observed in ME-extracted samples compared with the CNA kit. To validate the presumed augmented amplifiability of short-sized ctDNA fragments in ME-extracted ccfDNA, a mutation-specific ddPCR was performed on four ccfDNA samples that contained a mutation detectable with ddPCR (Appendix A). In all four cases, the number of mutant copies per mL of plasma was higher in ccfDNA extracted with the ME kit (Figure 6A), which was irrespective of the input amount of ccfDNA (Figure 6B). In addition, in all four samples a higher VAF was observed in ME-extracted ccfDNA compared with the CNA kit.

## 3. Discussion

In this study, the CNA kit served as the golden standard approach to evaluate the extraction of ccfDNA of other extraction methods regarding yield or integrity using plasmas of cancer patients, which eventually could lead to a higher sensitivity of variant detection. Based on our twenty-one plasma samples derived from cancer patients, both the integrity and levels of amplified short-sized fragments and tumor-specific mutants relative to the input amount of ccfDNA (in ng), as calculated for each reaction individually, revealed no differences between the RSC and CNA kits. However, when using the ccfDNA yield determined with DNA quantification methods such as Qubit or quantitative PCR approaches as reported in most other studies, the yield of ccfDNA, as well as the yield of short-sized fragments, is significantly higher using the CNA kit in cancer patient-derived plasmas. The ME kit seems a suitable methodology when extraction from high amounts of plasma is favored since, despite a lower yield per mL of plasma, higher mutant copy numbers and VAFs were observed. Since the use of different extraction methods might introduce bias to the mutation detection rate, we highly recommend applying the same ccfDNA extraction method within studies, especially when monitoring the treatment response based on multiple plasma samples, to prevent variation in mutant levels due to technical factors. This is further supported by our previous data on spiked-in samples [7]. 

The analysis of plasma-derived ccfDNA has recently become of importance in cancer diagnosis and treatment response monitoring. As ctDNA derives from the primary tumor or metastases, molecular characterization of the tumor is possible without the necessity to perform a tissue biopsy. Taking (multiple) biopsies is not always achievable and often accompanies health risks. With more targeted treatments available, the mutational status of the recurrent tumors or metastases has great implications for treatment decision making [30]. However, sample conditions should be optimal to be able to perform diagnostics on blood-based analytes and it is, therefore, key to critically analyze the plasma-derived material for accurate detection. Preanalytical procedures such as blood collection, transport, time before processing, and cell-free plasma processing have a major impact on the quality of the retrieved DNA and are the subject of several studies [24,25,31]. Furthermore, total ccfDNA concentrations can be influenced dramatically not only by technical factors such as hemolysis, but also by unrelated factors such as exercise and inflammation [32,33]. Finally, post-analytical determinants such as detection methods that differ in, e.g., sensitivity, complexity, and mutation coverage may also affect the clinical outcome [17,27,34,35].

In recent years, many kits have become commercially available for ccfDNA extraction [16,17]. Yield is the major outcome parameter when comparing different extraction methods. This study determined yields as measured by Qubit between 1.5 and 110 ng/mL of plasma, which overlaps with the range of ccfDNA generally found in both healthy individuals and cancer patients, affirming that assessing ccfDNA quality solely based on yield is challenging [10]. When using different ccfDNA extraction methods, the highest yield was obtained using the CNA kit, whereas the RSC kit was the least efficient. These observations are consistent with previous reports and, therefore, the CNA kit is considered as the gold standard reference approach when yield is the primary criterium [7,19,26]. 

To further analyze the integrity of the extraction product, the fragment size distribution and the amplifiability were tested. Two *ALU1* assays that target different sizes of DNA fragment lengths of 187 and 60 bp (Alu-187 and Alu-60) were used to compare the integrity index as determined by the Alu-187/Alu-60 ratio as reported previously [7]. This analysis revealed that the integrity index of ccfDNA of RSC was lower than that of CNA (data not shown), similar as reported for the spiked-in plasma. However, using the β-actin one-tube 3-size ddPCR assay, no significantly different 137/420 bp and 137/1950 fragments ratios were observed when comparing ccfDNA with the CNA and RSC kits. The Fragment Analyzer confirmed that the short-to-medium-sized fragments ratios are similar indicating that the integrity of the ccfDNA from the plasma of cancer patients does not differ between the CNA and RSC extraction methods. 

In agreement with the higher yield of ccfDNA as determined with Qubit when using CNA, the amplifiability of the 137 and 420 bp fragments using the β-actin one-tube 3-size ddPCR assay and using the CNA kit showed significantly more copies per mL of plasma compared to the RSC and Z kits. Previous data showed a similar reduced recovery and decreased number of shorter fragments of RSC-extracted ccfDNA compared to CNA [7,19]. However, these differences diminish in perspective of the number of copies per ng of ccfDNA, implying that the amplifiability of the short-sized fragments is similar when corrected for the input amount for both CNA and RSC kits in plasmas of cancer patients. The Z kit seems to preferentially extract long-sized fragments and was outperformed in respect to yield and amplifiability. 

The amplifiability of ctDNA was determined through the detection of tumor-specific mutations in ccfDNA in four samples. Interestingly, no differences could be detected regarding the number of mutant copies detected per mL of plasma, irrespective of ccfDNA input. However, RSC-extracted ccfDNA showed a higher VAF in three out of four cases. An increase in VAF when using the RSC kit for ccfDNA extraction is in agreement with the results we observed when using spiked-in plasma reported previously [7]. 

For multigene predictive testing using NGS a high input of ccfDNA is required for optimal and sensitive detection of ctDNA. Thus, processing high volumes of plasma in highly concentrated eluates is required [36]. Using six cancer patient-derived DLA samples, the numbers of copies of 137, 420, and 1950 bp fragments per mL of plasma were slightly (not significant) higher for all three fragment sizes with the CNA kit, while a higher total ccfDNA yield per mL of plasma was detected using the CNA kit as determined by Qubit. The number of copies of the 137 bp fragments per ng of input ccfDNA revealed a 2.2-fold increase compared to the CNA kit. Despite ccfDNA extracted with the ME kit revealed a relatively higher short-to-medium-sized fragment ratio as determined by the Fragment Analyzer, no significantly different 137/420 bp and 137/1950 bp fragments ratios were observed using the β-actin one-tube 3-size ddPCR assay. This PCR-based analysis indicated that the integrity of the ccfDNA from citrate plasma from cancer patients does not differ between the CNA and ME extraction methods in agreement with our analysis using spiked-in plasma samples [7]. A mutation-specific ddPCR assay to quantify ctDNA (i.e., tumor-specific mutants) levels in ccfDNA on four DLA samples revealed an increase in mutant copies per mL of plasma and elevated VAF in all ME-extracted ccfDNA samples compared to the CNA kit, irrespective of ccfDNA input. Thus, despite the lower total ccfDNA yield using the ME kit, relatively more ctDNA is extracted. The overall lower levels of the short-sized fragments recovered from citrate plasma samples compared to plasma samples collected in BCT tubes using the CNA kit (288 vs. 637 copies/mL of plasma, respectively) might be due to an abundance of long-sized fragments, as the short-to-medium fragment ratios determined with both the Fragment Analyzer and the β-actin one-tube 3-size ddPCR assay are lower for citrate plasma compared to the BCT plasma. However, as the citrate and BCT plasmas were not drawn from the same patients, factors such as stage of disease or response to therapy might explain these differences. Nevertheless, we have recently compared plasmas collected by DLA in citrate and from peripheral blood in Streck BCT-tubes from the same patients at the same time and using NGS analysis revealed a very high concordance between the VAFs of various tumor-specific variants [36].

Altogether, this study using cancer patient-derived plasmas shows that ccfDNA extraction using different extraction methods resulted in similar integrity and similar levels of amplified small-sized fragments and tumor-specific mutants per ng of ccfDNA input, but significant differences in the yield of ccfDNA as well as of small-sized fragments per mL of plasma, which makes alternating the application of different ccfDNA extraction methods lead to inconsistent results. The CNA kit consistently showed the highest yield of ccfDNA and of small-sized fragments, however, in the RSC kit higher VAFs were found, implying a preferential extraction of the mutation harboring ctDNA similar as observed in the artificial spiked-in plasma samples [7]. Recent studies showed that fragmentation of DNA in cell-free plasma differs between cancer patients and healthy individuals [5,37]. The average fragment size of ccfDNA is around the size of nucleosome-protected DNA (160–170 bp), while ctDNA in many cancers was shown to be 20–30 bp smaller (130–150 bp). Interestingly, in a cohort of 344 plasmas from 200 cancer patients, the analysis of the smaller size-selected ccfDNA fragments revealed clinically actionable mutations and copy number alterations at high frequency [37]. Although none of the commercially available kits are designed to enrich specifically for the smaller nucleosome bound ccfDNA fragments, Kloten et al. reported that extraction methods based on magnetic beads more efficiently recover short ccfDNA fragments compared to silico-based methods [23]. Since the number of cancer patient-derived plasma samples and DLA samples with a tumor-specific mutation is relatively low, additional studies are needed to confirm our observations.

Overall, these data suggest that the use of different extraction methods might introduce differences in the levels of mutant copies per mL of plasma and VAF due to technical factors, which might represent inaccurate discrepancies in clinical-relevant mutant copies crucial for clinical application, especially in treatment response monitoring. Therefore, continuous use of the same ccfDNA extraction method based on validated standard operating procedures is recommended to obtain comparable results. As long as there is no harmonization and standardization of procedures using preanalytical and analytical methods for liquid biopsy testing (e.g., primary diagnosis, minimal residual disease (MRD), response monitoring), it cannot yet be routinely implemented in the clinical setting [38].

## 4. Materials and Methods

### 4.1. Sample Collection and Processing

Plasma samples were collected from twenty-one patients with metastatic disease who were treated in the University Medical Center Groningen (UMCG, Groningen, The Netherlands) for GIST or NSCLC. GIST samples were collected in EDTA tubes (vacutainer #367525, Becton Dickinson, Franklin Lakes, NJ, USA), whereas NSCLC samples were collected in cell-free DNA blood collection tubes (BCTs) (Streck, Omaha, NE, USA). EDTA samples were processed within 4 h after venipuncture following guidelines as reported previously [39]. Samples were centrifuged for 10 min at 820× *g* to separate the lymphocytes from the plasma. The supernatant was centrifuged at 16,000× *g* for another 10 min to separate plasma from the remaining debris. The supernatants were transferred in 1 mL fractions and stored at −80 °C until ccfDNA extraction. Cell-free DNA BCTs were processed within 24 h using the same protocol except for a first centrifugation step at 1600× *g* following the manufacturer’s instructions. 

The DLA procedure was performed as previously described [40,41]. In short, procedures were performed on six patients with the Spectra Optia^®^ Apheresis System according to the standard continuous mononuclear cell (cMNC) protocol with a packing factor of 4.5 and the collection pump set to 1 mL per minute, hematocrit minus 3 percent points, and a flexible inlet flow and anticoagulation with anticoagulant citrate dextrose solution (starting concentration of 1:11). Following the cMNC protocol, up to 100 mL of plasma was collected with a packing factor of 20. The DLA samples were aliquoted and stored at −80 °C within 30 min after withdrawal. After thawing, DLA samples were centrifuged at 1600× *g* for 10 min to separate the plasma from the debris. All plasma processing was performed in a laboratory not used for any molecular testing to prevent contamination. For this validation study, the samples of patients with GIST were selected from a national GIST biobank study which is registered on ClinicalTrials.gov (NCT02331914), and the NSCLC samples from the lung plasma Biobank both at the UMCG. All patients gave written informed consent. Samples were selected irrespective of clinical and mutational status (Appendix A).

### 4.2. CcfDNA Extraction Techniques

In this study, four different methods for plasma ccfDNA extraction were used: CNA, RSC, Z, and ME (for specifications of each, see Table 1). CcfDNA was extracted from the same plasma sample using the CNA (0.9–2 mL), RSC (0.7–0.9 mL), and Z (0.8–0.9 mL) kits according to the corresponding manufacturer’s instructions. For the DLA samples, the same citrate plasma was used for ccfDNA extraction with CNA (2 mL) and ME (8 mL) according to the corresponding manufacturer’s instructions.

### 4.3. Yield and Integrity Assessment of the Different ccfDNA Extraction Methods

CcfDNA was quantified using the Qubit dsDNA HS assay kit on a Qubit 2.0 fluorometer (Thermo Fischer Scientific, Waltham, MA, USA). As a measure of integrity, we determined the fragment size distribution, integrity index, and amplifiability, similar as previously reported [7]. To determine fragment size distribution, the extracted ccfDNA samples were analyzed using the Fragment Analyzer. An amount of 2 µL of the ccfDNA samples was used for analysis according to the manufacturer’s instructions (Agilent, Santa Clara, CA, USA). A smear analysis for the 50–250 bp fraction was used as a representation of the short-sized fragments and the 250–450 bp fraction to represent the medium-sized fragments. The ccfDNA integrity index was assessed using two different *ALU1* PCR assays with lengths of 60 and 187 bp (TATAA, Göteborg, Sweden). The integrity index was calculated through the ratio between their quantitation cycle values (Alu-187/Alu-60) as previously reported [7]. The amplifiability of ccfDNA was evaluated using the β-actin one-tube 3-size ddPCR assay as described previously with minor adaptations [12]. In this multiplex assay, three different sized fragments of the β-actin gene are detected using the QX200™ Droplet Digital PCR System (Bio-Rad Laboratories, Hercules, CA, USA). Fragment sizes of 137 and 420 bp are detected with FAM- and HEX-labeled probes, respectively, and double positive droplets were counted as 1950 bp fragments. These primer and probe sequences were reported previously [42]. For mutation detection in ccfDNA extracted from plasma samples, mutation-specific ddPCR assays were performed. All applied primer and probe sequences are depicted in Appendix A. To demonstrate the ability to identify increased hemolysis, twelve plasma samples collected in EDTA tubes were stored for different times (4 h and five days) and at different temperatures (4 and 20 °C) prior to plasma processing (see Appendix A).

DdPCR assays for the β-actin one-tube 3-size ddPCR, KRAS G12/G13 screening, BRAF G466V, PDGFRA M844_D846del, TP53 H179R, TP53 R273H, and TP53 Y205C were performed using the Bio-Rad QX200™ platform included positive, wild type, and no template controls. Performance of the PCR setup was according to the manufacturer’s instructions. DdPCR analyses were performed on 5.0–8.8 µL of extracted plasma ccfDNA with different ccfDNA concentrations as measured by Qubit according to the manufacturer’s instructions. Briefly, the appropriate amounts of primer mix and probes for the β-actin one-tube 3-size ddPCR assay (1.1 µL of 4 µM 137 bp primer mix, 1.1 µL of 4 µM 420 bp primer mix, 1.1 µL of 6 µM 137 bp FAM labeled probe, and 1.1 µL of 6 µM 420 bp HEX labeled probe) and the mutation-specific ddPCR assays (Bio-Rad: 1.1 µL of primers and probe mix; IDT: 1.1 µL of primer mix, and 1.1 µL of probes) were added to 11 µL ddPCR supermix and supplemented with water when necessary up to a volume of 22 µL. All used primer and probe sequences are shown in Appendix A. Data were analyzed with the QuantaSoft^TM^ analytical software version 1.7.4.0917 and QuantaSoft^TM^ Analysis Pro 1.0.596 (both Bio-Rad). Positive, wild type, and no template controls were used to establish cutoff levels. Droplet counts were used to calculate the number of copies per initial volume of plasma input as well as the VAF calculated by the QuantaSoft^TM^ Analysis Pro 1.0.596 software. All molecular testing was performed in the ISO15189-accredited laboratory of molecular pathology at the UMCG. All standard precautions were taken to avoid contamination of amplification products using separate laboratories for pre- and post-PCR handling.

### 4.4. Statistical Analyses

Statistical analyses were performed using the IBM SPSS statistics version 25.0 (IBM, Armonk, NY, USA), R (version 3.6.0) and R Studio software (R Studio, Boston, MA, USA), and Prism 7.0 (GraphPad software, San Diego, CA, USA). For statistical assessment, a one-way ANOVA with repeated nonparametric measures (Friedman test) was performed followed by a Dunn’s multiple comparison test. In case there were only two paired samples, a generalized linear mixed model was applied.

## 5. Conclusions

When using cancer patient-derived ccfDNA from blood plasma instead of artificial spiked-in reference samples, preanalytical conditions significantly influence the overall result of a ccfDNA extraction method. Inconsistent processing of the plasma and the use of different ccfDNA extraction kits might contribute to incorrect results, which eventually can lead to inappropriate variant calling or inaccurate VAF determination. Whatever ccfDNA extraction kit is selected will be up to personal preferences, however, it should not be changed within a cohort in order to preserve similar conditions in all cases. For biobanking of liquid biopsies, our findings also recommend the storage of cell-free plasma and not of extracted ccfDNA. Harmonization of procedures using preanalytical conditions will strongly improve interstudy similarity and compatibility and thereby contribute to the implementation of liquid biopsy approaches in the clinical practice.

## Figures and Tables

**Figure 1 cancers-12-01222-f001:**
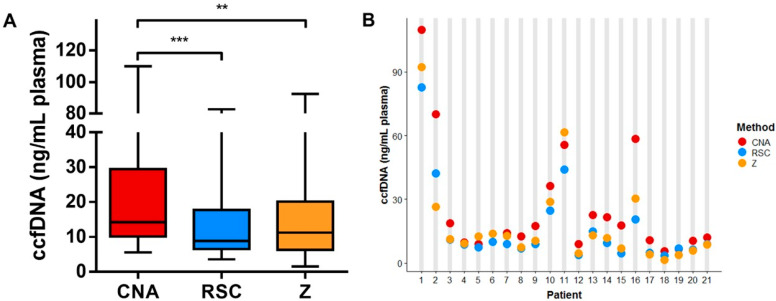
Comparison of the circulating cell-free DNA (ccfDNA) yield among different extraction methods. (**A**) Boxplot illustrating the yield as measured with Qubit per method. Horizontal lines represent the median, the boxes, and the interquartile range. (**B**) Individual patient ccfDNA yields for the different extraction methods are displayed for the plasma of twenty-one nonsmall cell lung carcinoma (NSCLC) patients. One-way ANOVA multiple comparison test, ** *p* < 0.01, *** *p* < 0.001.

**Figure 2 cancers-12-01222-f002:**
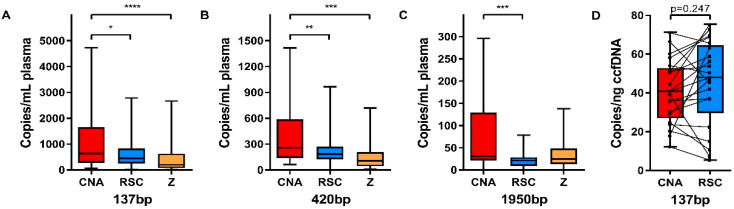
Comparison of β-actin fragment sizes among different ccfDNA extraction kits. Boxplots illustrating the number of copies per mL of plasma for the 137 (**A**), 420 (**B**), and 1950 bp (**C**) fragment sizes, as well as the 137 bp copies per ng of ccfDNA (**D**) as measured with the one-tube 3-sized β-actin ddPCR assay. Horizontal lines represent the median, the boxes, and the interquartile range. All median values and interquartile ranges are depicted in Appendix A. One-way ANOVA multiple comparison test, * *p* < 0.05, ** *p* < 0.01, *** *p* < 0.001, **** *p* < 0.0001.

**Figure 3 cancers-12-01222-f003:**
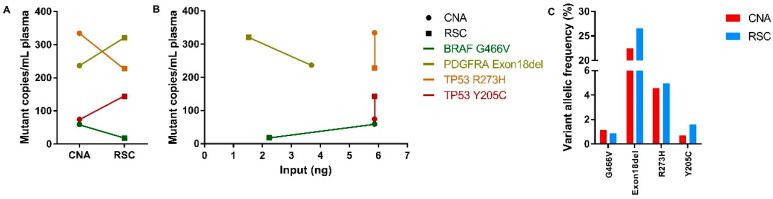
Mutation detection in ccfDNA extracted from plasma with the QIAamp Circulating Nucleic Acid (CNA) and Maxwell RSC ccfDNA Plasma (RSC) methods. (**A**) Before-after plot illustrating the number of mutant copies per mL of plasma for four paired CNA-extracted (dots) and RSC-extracted (squares) samples. (**B**) XY plot illustrating the detected mutant copies per mL of plasma plotted against the input in ng on the x-axis. Similarly colored results originate from the same patient harboring a detectable mutation according to the legend. (**C**) Bar graph illustrating the detected higher variant allelic frequency (VAF) in a percentage of four paired CNA-extracted (red) and RSC-extracted (blue) samples.

**Figure 4 cancers-12-01222-f004:**
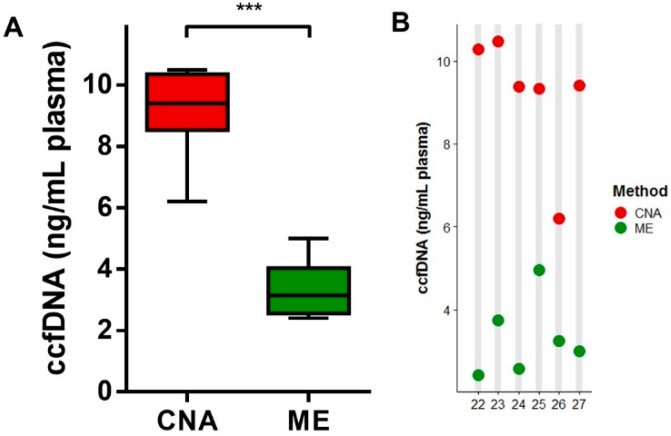
Comparison of the ccfDNA yield using the CNA and ME extraction methods in diagnostic leukapheresis (DLA) samples. (**A**) Boxplot illustrating the yield as measured with Qubit per method. Horizontal lines represent the median, the boxes, and the interquartile range. (**B**) Individual patient ccfDNA yields for the different extraction methods are displayed for the citrate plasma of six NSCLC patients. Generalized linear mixed model, *** *p* < 0.001.

**Figure 5 cancers-12-01222-f005:**
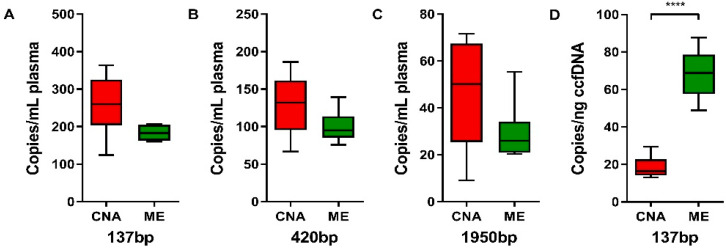
Comparison of β-actin fragment sizes among the CNA and ME extraction methods in DLA samples. Boxplots illustrating the number of copies per mL of plasma for the 137 (**A**), 420 (**B**), and 1950 bp (**C**) fragment sizes, as well as the 137 bp copies per ng of ccfDNA (**D**) as measured with the one-tube 3-sized β-actin ddPCR assay. Horizontal lines represent the median, the boxes, and the interquartile range. All median values and interquartile ranges are depicted in Appendix A. Generalized linear mixed model, * *p* < 0.05, *** *p* < 0.001.

**Figure 6 cancers-12-01222-f006:**
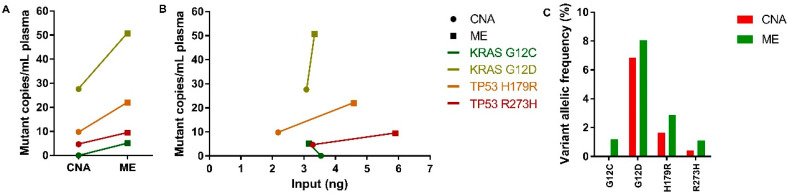
Mutation detection among the CNA and ME extraction methods in DLA samples. (**A**) Before-after plot illustrating the number of mutant copies per mL of plasma for four paired CNA-extracted (dots) and ME-extracted (squares) samples. (**B**) XY plot illustrating the detected mutant copies per mL of plasma plotted against the input in ng on the x-axis. Similarly colored results originate from the same patient harboring a detectable mutation according to the legend. (**C**) Bar graph illustrating the detected VAF in a percentage of four paired CNA-extracted (red) and ME-extracted (green) samples.

**Table 1 cancers-12-01222-t001:** Specifications of the ccfDNA extraction kits used in this study.

Kit	Manufacturer	Method	Input Volume (mL)	Elution Volume (µL)	Execution
QIAamp Circulating Nucleic Acid Kit (CNA)	Qiagen	Silica-based	1–4 (0.9–2)	20–150 (47–92)	Manual
Maxwell RSC ccfDNA Plasma Kit (RSC)	Promega	Magnetic beads	0.2–1 (0.7–0.9)	50 (50)	Automated
Zymo Quick ccfDNA Serum & Plasma Kit (Z)	Zymo research (BaseClear)	Silica-based	<10 (0.8–0.9)	>50 (47)	Manual
QIAamp MinElute ccfDNA midi kit (ME)	Qiagen	Magnetic beads	4–10 (8)	20–80 (47)	Manual

Specifications of different kits. The amounts that were used for this study are displayed within brackets. For detailed information regarding the kit specifications, we recommend accessing the websites of the manufacturers.

**Table 2 cancers-12-01222-t002:** Short- and medium-sized fragment percentages of citrate plasma as measured with the Fragment Analyzer.

	Fragment Analyzer	β-Actin One-Tube 3-Size ddPCR	
Kit	Ratio S/M	Ratio 137/420 bp	Ratio 137/1950 bp
CNA	1.81 (1.58–2.67)	1.64 (1.54–1.95)	6.56 (5.55–10.3)
ME	3.10 (2.37–3.76) *	1.73 (1.48–1.89)	8.00 (6.47–9.31)

Measurements are displayed as a median percentage of retrieved fragment size with the interquartile range within brackets. Ratio S/M: Ratio between short-sized fragments (50–250 bp) and medium-sized fragments (250–450 bp). * *p* < 0.05 between the CNA and QIAamp MinElute ccfDNA (ME) extraction methods.

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
