# Peer review of "Comparison of Circulating Cell-Free DNA Extraction Methods for Downstream Analysis in Cancer Patients"

_cancers, 2020, doi:10.3390/cancers12051222_

Round 1
Reviewer 1 Report
In this manuscript, Leest et al compared circulating cell-free DNA extraction methods for cancer patients with commercially available kits. The manuscript fits the scope of Cancers, but there are not any newly developed ccfDNA extraction methods and authors performed ccDNA extraction by commercially available kits. They just compared these kits, and this work will be more appropriate for consideration by other methodology journals.
Author Response
Reviewer 1: Comments and Suggestions for Authors
In this manuscript, Leest et al compared circulating cell-free DNA extraction methods for cancer patients with commercially available kits. The manuscript fits the scope of Cancers, but there are not any newly developed ccfDNA extraction methods and authors performed ccfDNA extraction by commercially available kits. They just compared these kits, and this work will be more appropriate for consideration by other methodology journals.
Answers to comments of reviewer 1:
Our group (M. Ti. and E.S.) was involved in the recently published multicenter evaluation of workflows focusing on different ccfDNA extraction methods by the CANCER-ID consortium (Lampignano et al. Multicenter Evaluation of Circulating Cell-Free DNA Extraction and Downstream Analyses for the Development of Standardized (Pre)analytical Work Flows. Clin. Chem. Oct. 2019). This CANCER-ID study revealed considerable differences between various ccfDNA extraction methods regarding the quantity and integrity of extracted ccfDNA. For this multicenter study, artificial spiked-in plasma was used and this analysis showed the relevance of carefully selecting extraction methods and considering preanalytical conditions of the extracted ccfDNA. In line with this CANCER-ID study, the aim of the present study was to evaluate the quantity and integrity of extracted ccfDNA from cancer patient-derived plasma samples (in contrast to artificial spiked-in pooled plasma samples) using different ccfDNA extraction kits including the QIAamp Circulating Nucleic Acid Kit (CNA) and Maxwell RSC ccfDNA Plasma Kit (RSC) that showed significant differences in the multicenter CANCER-ID study.
To clarify the aim of our study, we changed the first paragraph of the abstract (lines 18-24) into: “Circulating cell-free DNA (ccfDNA) may contain DNA originating from the tumor in plasma of cancer patients (ctDNA) and enables non-invasive cancer diagnosis, treatment predictive testing and response monitoring. A recent multicenter evaluation of workflows by the CANCER-ID consortium using artificial spiked-in plasma showed significant differences and consequently the importance of carefully selecting ccfDNA extraction methods. Here, the quantity and integrity of extracted ccfDNA from plasma of cancer patients were assessed.”
And in the introduction (lines 83-94) into: “A recent multicenter evaluation of workflows by the CANCER-ID consortium revealed considerable differences between various ccfDNA extraction methods regarding the quantity and integrity of extracted ccfDNA using artificial spiked-in plasma and showed the relevance of carefully selecting extraction methods and considering preanalytical conditions of the extracted ccfDNA [7]. In line with this CANCER-ID study, the aim was to evaluate the quantity and integrity of extracted ccfDNA from cancer patient-derived plasma samples using different ccfDNA extraction kits. Plasma samples from patients with either a gastrointestinal stromal tumor (GIST) or non-small cell lung carcinoma (NSCLC) were selected for the comparison of three different plasma ccfDNA extraction techniques (QIAamp Circulating Nucleic Acid Kit (CNA), Maxwell RSC ccfDNA Plasma Kit (RSC), and Zymo Quick ccfDNA Serum & Plasma Kit (Z)).”
Reviewer 2 Report
In this manuscript, four commercial kits for circulating cell-free DNA (ccfDNA) extraction were compared with respect to DNA yield, integrity, and amplifiability. Based on the found differences, the authors particularly concluded that standardization of ccfDNA extraction is necessary for comparability of clinical testing both within and between studies.
Cancer diagnosis using ccfDNA analysis is minimally invasive and could simplify sample preparation, particularly in treatment response monitoring. CcfDNA extraction is a crucial step in this analysis. Several papers comparing ccfDNA extraction have already been published. The major contribution of this manuscript is in utilization of natural plasma samples from cancer patients and careful evaluation of preanalytical conditions of the extracted ccfDNA.
Specific comments:
The “VAF” abbreviation should be specified in the Abstract.
In Table 1, column width should be modified (some words are divided into two lines).
In Figure 5, graphs of Copies/ng ccfDNA for 420bp and 1950bp fragments could be added.
Line 226: How did you determine the DNA yield “in ng”? Was it different from the yield specified in the next sentence?
Line 308: Please correct “between de VAF”.
Lines 315-316: Do you have any explanation for “a preferential extraction of the mutation harboring ctDNA.”
Lines 381-382: Please correct “… at temperatures (4°C and 20°C) plasma processing“.
Reviewer 3 Report
To what extent is average fragment size determined by the proportion of normal ccfDNA versus ctDNA as opposed to DNA degradation pre and during processing. Are the fragments sizes clustered or continuous (wrt conjecture about the nature and organ of the different types of DNA present)?
Given that the end game is to detect variants that may have significance for treatment choices and outcome, the number of samples analysed for variants is small.
Author Response
Reviewer 3: Comments and Suggestions for Authors
Comment: To what extent is average fragment size determined by the proportion of normal ccfDNA versus ctDNA as opposed to DNA degradation pre and during processing. Are the fragments sizes clustered or continuous (wrt conjecture about the nature and organ of the different types of DNA present)?
Answer to comment of reviewer 3:
Indeed, recent studies showed that fragmentation of DNA in cell free plasma differs between cancer patients and healthy individuals (Mouliere et al., Sci. Transl. Med. 10, eaat4921 (2018)). The average fragment size of ccfDNA is around the size of nucleosome-protected DNA (160-170bp), while ctDNA was shown to be 10 or 20 bp smaller (130-150bp). Most interesting, the analysis of size-selected cfDNA identified clinically actionable mutations and copy number alterations in a cohort of 344 plasmas from 200 cancer patients. Today ccfDNA extraction techniques are unable to distinguish between fragments with these small differences.
We have rephrased the text in the discussion (lines 318-326) into: “The CNA kit consistently showed the highest yield of ccfDNA and of small-sized fragments, however in the RSC kit higher VAFs were found, implying a preferential extraction of the mutation harboring ctDNA similar as observed in the artificial spiked-in plasma samples [7]. Recent studies showed that fragmentation of DNA in cell-free plasma differs between cancer patients and healthy individuals [5,37]. The average fragment size of ccfDNA is around the size of nucleosome-protected DNA (160-170bp), while ctDNA in many cancers was shown to be 20-30bp smaller (130-150bp). Interestingly, in a cohort of 344 plasmas from 200 cancer patients, the analysis of the smaller size-selected ccfDNA fragments revealed clinically actionable mutations and copy number alterations at high frequency [37].”
In general, degraded DNA during processing is very small in size (<100bp) and overall we do not observe degraded ccfDNA in our plasmas using analysis with the fragment bioanalyzer. However, since all samples used in this study were processed from the same batch of plasma, and the same plasmas from the same batches were used for the different ccfDNA extraction methods, the degradation conditions were considered equal. Besides, small-sized degraded artifacts would not interfere with the ddPCR analysis, which only detects fragments larger than 137bp.
Comment: Given that the end game is to detect variants that may have significance for treatment choices and outcome, the number of samples analysed for variants is small.
Answer to comment of reviewer 3:
Our group (M. Ti. and E.S.) was involved in the recently published multicenter evaluation of workflows focusing on different ccfDNA extraction methods by the CANCER-ID consortium (Lampignano et al. Multicenter Evaluation of Circulating Cell-Free DNA Extraction and Downstream Analyses for the Development of Standardized (Pre)analytical Work Flows. Clin. Chem. Oct. 2019). This CANCER-ID study revealed considerable differences between various ccfDNA extraction methods regarding the quantity and integrity of extracted ccfDNA. For this multicenter study, artificial spiked-in plasma was used and this analysis showed the relevance of carefully selecting extraction methods and considering preanalytical conditions of the extracted ccfDNA. In line with this CANCER-ID study, the aim of the present study was to evaluate the quantity and integrity of extracted ccfDNA from cancer patient-derived plasma samples (in contrast to artificial spiked-in pooled plasma samples) using different ccfDNA extraction kits including the QIAamp Circulating Nucleic Acid Kit (CNA) and Maxwell RSC ccfDNA Plasma Kit (RSC) that showed significant differences in the multicenter CANCER-ID study. To our opinion, 21 different cancer-derived plasma samples should be sufficient to conclude that (lines 332-337): “… these data suggest that the use of different extraction methods might introduce differences in the levels of mutant copies per mL of plasma and VAF due to technical factors, which might represent inaccurate discrepancies in clinical-relevant mutant copies crucial for clinical application especially in treatment response monitoring. Therefore, continuous use of the same ccfDNA extraction method based on validated standard operating procedures are recommended to obtain comparable results.”
The collection of sufficient plasma from the same patients for research purposes is already a challenge especially from patients at tlate-stage of disease, and is a reason for us to restrict the analysis to 3 ccfDNA extraction methods only using those plasma samples in the biobank with more than 4ml. To clarify the aim of our study, we changed the introduction (lines 83-94) on this point.
However, as the number of samples with a tumor-specific mutation to study the extraction of ctDNA versus ccfDNA is low, we agree that independent confirmation of these data is in place. To address this in our paper, we added the following sentence in the discussion (lines 330-331): “Because the number of cancer patient-derived plasma samples and DLA samples with tumor-specific mutation is relatively low, additional studies as needed to confirm our observations.”
Round 2
Reviewer 1 Report
I have understood the aim of the manuscript and the difference between previous (Clin. Chem., 2019) and this studies. Thank you again for clarifying the aim of the manuscript and kind response to the comment.
Minor issues
- Needed for editing Fig S1 to be readable easily (X-, Y-axis, color bar legends in graphs).
2. Needed for correction of Table S4 (missing right part of the table).